# Evaluation of the Potential Risk Posed by Emerging *Yr5*-Virulent and Predominant Races of *Puccinia striiformis* f. sp. *tritici* on Bread Wheat (*Triticum aestivum* L.) Varieties Grown in Türkiye

**DOI:** 10.3390/jof11090635

**Published:** 2025-08-29

**Authors:** Kadir Akan, Ahmet Cat, Medine Yurduseven, Yesim Sila Tekin, Mehmet Zahit Yeken, Mehmet Tekin

**Affiliations:** 1Department of Plant Protection, Faculty of Agriculture, Kırşehir Ahi Evran University, 40100 Kırşehir, Türkiye; kadir_akan@hotmail.com; 2Department of Plant Protection, Faculty of Agriculture, Siirt University, 56100 Siirt, Türkiye; ahmetcat@siirt.edu.tr; 3Department of Field Crops, Institute of Natural and Applied Sciences, Akdeniz University, 07070 Antalya, Türkiye; yurdusevenmdn07@gmail.com (M.Y.); ysilatekin@gmail.com (Y.S.T.); 4Argeto Vegetable Seeds Co., 07112 Antalya, Türkiye; 5Merkez Anadolu Kimya Co., 07190 Antalya, Türkiye; 6Department of Field Crops, Faculty of Agriculture, Bolu Abant İzzet Baysal University, 14030 Bolu, Türkiye; yekenmehmetzahit@gmail.com; 7Department of Field Crops, Faculty of Agriculture, Akdeniz University, 07070 Antalya, Türkiye

**Keywords:** bread wheat, *Puccinia striiformis* f. sp. *tritici*, resistance, stripe rust

## Abstract

In this study, the reactions of 70 bread wheat varieties released in Türkiye to five prevalent *Pst* races, including the *Yr5*-virulent PSTr-27, were evaluated. Reaction tests of wheat varieties to all races revealed PSTr-27 as the most aggressive race, followed by PSTr-31, PSTr-28, PSTr-29, and PSTr-30. Notably, only seven varieties (Kıraç 66, İkizce 96, Dinç, Altındane, Ziyabey 98, Bayraktar 2000, and Shiro) exhibited moderately resistant reactions to PSTr-27, while the remaining varieties were susceptible. The presence of nine important resistance (*Yr*) genes in these varieties was also screened at the molecular level. *Yr5*, *Yr15*, and *Yr26* genes were not detected in any of the varieties and *Yr10* and *YrSP* genes were each detected in only one variety, while the other genes were detected in different ratios. Molecular screening showed that 19 varieties with no resistance genes used in this study displayed susceptible reactions; however, ten varieties that did not carry any resistance genes showed resistant reactions to one or more races, suggesting the presence of unknown or novel resistance sources. Furthermore, gene combinations, particularly *Yr10* + *Yr18*, significantly provided resistance to all *Pst* races studied. These findings highlight that continual monitoring of PSTr-27, and other *Pst* races is needed, since it can be a serious threat to wheat production in Türkiye and neighboring countries.

## 1. Introduction

Wheat (*Triticum* L.) is one of the most important cereals worldwide and ranks third in production, followed by maize and rice. Global wheat production in 2023 was approximately 800 million tons [1]. Wheat is also the most strategic crop in Türkiye, and the country is one of the largest producers of wheat globally. At the same time, Türkiye is one of the leading countries in the consumption of wheat products, with approximately 160 kg per capita [1]. However, wheat production is threatened by many diseases both in Türkiye and worldwide. Among them, wheat stripe rust, caused by *Puccinia striiformis* f. sp. *tritici* (*Pst*), is the most devastating disease. This disease has been reported in at least 60 countries, and it is known that 88% of global wheat-growing areas are susceptible to this disease [2]. Epidemics caused by this disease have been known to occur regularly every two years, affecting more than 25% of wheat-growing areas in many European, North African, and Middle Eastern countries, including Türkiye [3].

Historically, epidemics have occurred at varying intensities throughout Türkiye, depending on weather conditions and host resistance [4,5,6]. Although several epidemics were recorded in the inner regions of the country before the 1960s, yield losses of up to 80% were reported in some areas between the 1960s and 1990s [7]. Between the 1990s and 2010s, *Pst* epidemics lead to dramatic yield and quality losses [5,8]. Despite these epidemics, studies focusing on race detection causing the epidemics were very few until recent years. In these studies, resistance genes such as *Yr2*, *Yr6*, *Yr7*, and *Yr9* were determined to be ineffective against *Pst* populations in the country [5,9,10]. Additionally, the differential lines used in these studies for race detection commonly include genotypes with more than one *Yr* gene. However, it is known that using near isogenic lines (NILs) with the same genetic background but each containing different resistance genes provides more reliable results. The use of such differentials also facilitates global standardization when detecting races accurately. Recently, with the use of Avocet NILs, developed by Wellings et al. [11], Cheng and Chen [12] and Wan and Chen [13], 25 races were identified in the coastal regions of Türkiye [14], and 38 races were detected throughout the country [6]. The most frequently detected races are *PSTr-29*, *PSTr-30*, *PSTr-31*, and *PSTr-28*, respectively. The virulence formula of *PSTr-29* is identical to that of *PSTv-36*, which was reported in the USA by Wan and Chen [13] and Chen et al. [15], exemplifying the intercontinental migration of races.

Until now, over 80 resistance genes and many QTLs have been identified against *Pst* [16]. Among them, *Yr5* and *Yr15* confer resistance to *Pst* races worldwide. Although no races virulent to *Yr15* have been reported, several cases of virulence to *Yr5* have been reported, particularly in recent years. One such case was reported in Türkiye [6,17], with others reported in India [18], Australia [19], China [20], and Syria [21].

The virulence to *Yr5* has been reported in China, Turkey and Syria over the last five years. This indicates that a virulent race or other variants to *Yr5* have spread or evolved spontaneously. Therefore, studying virulence to *Yr5* is a high-priority issue, both strategically and economically, particularly in regions such as Türkiye and Syria where primary/alternate host populations, including *Berberis* spp. [22,23] and wild relatives of wheat [24], are prevalent. For example, the evolution of a race or races virulent to *Yr9*, followed by their spread across Asia, caused epidemics that led to significant yield losses in the 1990s and early 2000s. During those years, it was estimated that the stripe rust-prone areas were mainly in China (9.6 m hectares), India (9.4 m hectares), Türkiye (7.4 m hectares), Pakistan (5.8 m hectares), Iran (4.4 m hectares), and Syria (1.3 m hectares) [25]. Therefore, preliminary studies should be conducted to assess the potential risks of *Yr5*-virulent races to wheat production. The potential risk of *Yr5*-virulent races in China was evaluated on 165 Chinese wheat cultivars. It was reported that a low percentage of these cultivars have postulated resistance genes *Yr5*, *Yr7*, *Yr10*, *Yr15*, *Yr26*, or *YrSP* and the isolates TSA-6 and TSA-9, which are virulent to *Yr5*, can be a serious threat to wheat production [26].

This study aimed to evaluate the potential risk of the *Yr5*-virulent and other predominant races previously reported in studies on bread wheat production in Türkiye. For this purpose, reaction tests were performed to evaluate the virulence of *Yr5*-virulent race in comparison with other prevalent races, and molecular detection of *Yr5* along with other important *Yr* genes in bread wheat varieties was carried out using molecular markers.

## 2. Materials and Methods

### 2.1. Genetic Materials

A total of 70 bread wheat (*T*. *aestivum*) varieties, released from 1968 to 2022 in Türkiye, were used as the plant material (Appendix A). Some of these varieties (especially the older ones, based on year of registration) have been widely cultivated for many years, while the rest have been newly registered. In addition, the wheat variety Morocco, susceptible to all known *Pst* races, was used both as a susceptible control and to multiply urediniospores of each race.

A set of 16 *Yr* NILs with the Avocet S background were used as differentials to determine the virulence (Vr) and avirulence (Avr) patterns of the races. Reactions of the 70 wheat varieties were evaluated with five *Pst* races, *PSTr-27* Vr to *Yr5* (Vr: *Yr5*,*6*,*7*,*17*,*24*,*27*,*44*,*SP*/Avr: *Yr1*,*8*,*9*,*10*,*15*,*32*,*43*,*Tr1*,*Exp2*,*Tye*) and *PSTr-28* (Vr: *Yr6*,*7*,*8*,*9*,*24*,*27*,*43*,*44/Avr: Yr1*,*5*,*10*,*15*,*17*,*32*,*SP*,*Tr1*,*Exp2*,*Tye*), *PSTr-29* (Vr: *Yr6*,*7*,*8*,*9*,*27*,*43*,*44*,*Tr1*,*Exp2/Avr: Yr1*,*5*,*10*,*15*,*17*,*24*,*32*,*SP*,*Tye*), *PSTr-30* (Vr: *Yr6*,*8*,*9*,*43*,*44*,*Tr1*,*Exp2*,*Tye/Avr: Yr1*,*5*,*10*,*17*,*24*,*27*,*32*) and *PSTr-31* (Vr: *Yr1*,*6*,*7*,*9*,*17*,*27*,*32*,*43*,*44*,*Tr1*,*Exp2*/Avr: *Yr5*,*8*,*10*,*15*,*24*,*Tye*), currently prevalent and Avr to *Yr5* [6].

### 2.2. Multiplication of Pst Urediniospores and Reaction Tests of the Varieties

Urediniospores of each race were multiplied using the method described by Cat et al. [6]. Seeds of the susceptible check variety Morocco were sown in small plastic pots filled with peat substrates (TS1, Klassman GmbH, Geeste, Germany). The pots were placed in trays and transferred to a growth chamber. Fourteen-day old seedlings at the two-leaf stage were inoculated with urediniospores of each race.

The suspension of fresh urediniospores was prepared by adding 10 mg of urediniospores to 5 mL of hydrofluoroether (Novec^TM^ 7100, 3M Co., St. Paul, MN, USA) in a 25 mL of atomizer of airbrush spray gun, and then each suspension was sprayed onto the seedlings as described by Sorensen et al. [27]. The inoculated seedlings were then incubated at 10 °C for 24 h at 100% relative humidity (RH) in darkness. After incubation, the plants were transferred to climate-controlled rooms programmed with a diurnal temperature cycle (10–16 °C) with an 8 h dark/16 h light period, and 60% relative humidity. Each pot was enclosed in a cellophane bag to prevent cross-contamination. Approximately 19 days post-inoculation (dpi), fresh urediniospores of each race were collected into gelatin capsules using a mini cyclone collector. The multiplication of urediniospores on the susceptible variety was repeated as necessary until enough spores were obtained for reaction tests.

Following the multiplication of urediniospores, reaction tests were conducted on 70 bread wheat varieties used in the study. For this purpose, 10 seeds of each variety were sown in small plastic pots. The same methods for inoculation, incubation, and cultivation described in the multiplication of urediniospores section were applied in the reaction tests [27]. The testing was conducted with three replications to confirm the accuracy of the variety resistance to *Pst* races. Infection type (IT) for each variety was recorded 19 dpi using the 0–9 scale developed by McNeal et al. [28]. On this scale, infected plants scored as ITs 0–3 were considered resistant (R) (necrotic/chlorotic blotches with no or trace sporulation); ITs 4–6 were considered moderate-resistant/susceptible (necrotic/chlorotic blotches with moderate sporulation); and ITs 7–9 were considered susceptible (high sporulation with no or little necrotic/chlorotic blotches).

### 2.3. Extraction of Genomic DNA

Genomic DNA was extracted from approximately 100 mg of non-infected fresh leaves collected from fourteen-day-old seedlings of the tested varieties, the check variety in Morocco, and differential lines each carrying a different *Yr* resistance gene at the two-leaf stage. Leaf samples and lysis buffer were first added to a 2.0 mL microcentrifuge tube, and the mixture was homogenized using a micro pistil and vortexed regularly. The following procedures were performed in accordance with the manufacturer’s protocol for the NucleoSpin^®^ Plant II Extraction Kit (Macherey-Nagel, Dueren, Germany). Genomic DNA was dissolved in 100 μL of Tris-EDTA (TE) buffer, and the quality and concentration were assessed with 1% agarose gel electrophoresis with a DNA standard. It was diluted with Tris-EDTA (TE) buffer (pH 8.0) to adjust a final concentration of 50 ng/μL for PCR amplification.

### 2.4. Molecular Detection of Yr Genes

Molecular detection was performed using PCR amplification with different molecular markers linked to the resistance genes *Yr5*, *Yr10*, *Yr15*, *Yr17*, *Yr18*, *Yr26*, *Yr36*, *Yr44,* and *YrSP* to determine the presence/absence of these genes in the varieties. Information about these markers was given in Table 1. The genomic DNAs of differential lines, each carrying a different *Yr* resistance gene, were used as positive control. The total volume of PCR reaction mixture was 20 uL, containing 50 ng DNA template, 1X PCR buffer (Thermo Fisher Scientific, Waltham, MA, USA), 1.5 mM MgCI2 (Thermo Fisher Scientific, Waltham, MA, USA), 0.2 mM of dNTPs (Thermo Fisher Scientific, Waltham, MA, USA), 1 μM forward primer, 1 μM reverse primer, and 1 U Taq DNA polymerase (Thermo Fisher Scientific, Waltham, MA, USA). Amplifications were performed in a thermal cycler (T100, Bio-Rad, Hercules, CA, USA) under the following conditions: initial denaturation at 94 °C for 5 min, followed by 35 cycles of denaturation at 94 °C for 30 s, annealing at 45–65 °C (Table 1) for 30 s, and extension at 72 °C for 1 min, and a final extension of 10 min at 72 °C, before cooling down to 4 °C. PCR products were electrophoresed with a 2% (*w*/*v*) agarose gel in 1× TBE (Tris-boric acid-EDTA) buffer (pH 8.3) in a horizontal electrophoresis system (Bio-Rad, Hercules, CA, USA) at 5 V/cm for 40 to 60 min. DNA standards of 50 bp and 100 bp (Thermo Fisher Scientific, Waltham, MA, USA) were used to determine the size of amplicons. Visualization of gels was conducted under UV light using a gel imaging system (UVsolo touch, Analytik Jena, Jena, Germany) following staining with ethidium bromide.

### 2.5. Data Analysis

A 0–9 scale was used to determine the reaction of the varieties to *Pst* races, with three replications for each test. The resulting data were recorded in Microsoft Excel. The basic statistics such as the mean, standard deviation, coefficient of variation (CV), standard deviation (SD), kurtosis, and skewness were first calculated. Additionally, standard errors were used to compute z values for reactions to each *Pst* race, and *p*-values were calculated to indicate the degree of deviation from normal distribution for skewness and kurtosis values. To assess the effect of each *Yr* gene or gene combination in various genetic backgrounds, the data were divided into two groups based on presence or absence of related genes. A one-way analysis of variance (ANOVA) was performed at a 95% confidence level to identify significant differences between these groups. All analyses were performed in an R environment.

## 3. Results

### 3.1. Reaction Tests of the Bread Wheat Varieties to Pst Races

Reactions of all varieties to the prevalent *Pst* races, *PSTr-28*, *PSTr-29*, *PSTr-30*, and *PSTr-31* in Türkiye, as well as to the *PSTr-27*, a virulent race to *Yr5*, were determined. The basic statistics and generated histograms of the reactions of all varieties to the *Pst* races are given in Appendix A and Figure 1. Overall, it was determined that the most virulent race among the varieties tested was *PSTr-27* (IT: 7.67), as expected. This was followed by *PSTr-31* (7.27), *PSTr-29* (6.63), *PSTr-28* (6.61) and *PSTr-30* (6.61), respectively. While *PSTr-28* and *PSTr-29* had the highest CV value with 16.54% and 15.66%, *PSTr-27* had the lowest with 11.92% (Appendix A). Additionally, the skewness and kurtosis of the reactions of the varieties to *PSTr-28* were close to zero, indicating that the distribution of reactions among the varieties was fairly consistent (Appendix A; Figure 1).

The results of the reaction analyses of the varieties to all *Pst* races are presented in Table 2. As mentioned above, the most virulent race was identified as *PSTr-27*, and except for seven varieties (Kıraç 66, İkizce 96, Ziyabey 98, Bayraktar 2000, Altındane, Dinç and Shiro) that showed a moderately resistant (MR) reaction, no variety showed a resistant (R) reaction to this race. According to the scale, 19 varieties scored seven, 28 varieties scored eight, and 16 varieties scored nine, all indicating a susceptible (S) reaction to this race. Against the second most virulent race, *PSTr-31*, the varieties were scored between four and eight; while seven varieties were identified as moderately resistant (MR), the remaining 63 had a susceptible (S) reaction. For the third most virulent race, *PSTr-29*, only one variety (Kıraç-66) exhibited a resistant (R) reaction. In addition, 27 varieties were moderately resistant whereas 42 varieties showed a susceptible reaction. *PSTr-28* and *PSTr-30* were determined to have similar virulence on average on the varieties. While only one variety, Dinç, showed resistant reaction to *PSTr-30*, 28 varieties had moderately resistant reaction. The remaining 41 varieties were scored with seven and eight, indicating susceptible reaction. There was no variety with resistant reaction to *PSTr-28*; however, a similar variability was observed in this race as in the *PSTr-30*. While 27 varieties had a moderately resistant reaction to *PSTr-28*, the remaining 43 varieties were susceptible (Table 2).

Overall, Kıraç 66 and Dinç were the most prominent varieties with an MR or R reaction to all races including *Yr5*-virulent, *PSTr-27*. In addition, some varieties displayed resistance to more than one race (Table 3). Ziyabey 98, Bayraktar 2000, and Shiro were the most prominent varieties with moderately resistant reaction to *Yr5*-virulent race, *PSTr-27*, and another three races: *PSTr-28*, *PSTr-29*, and *PSTr-30*. The other prominent varieties, Dağdaş 94, Gönen 98, and Genesi, were resistant or moderately resistant to all races except for *PSTr-27* (Table 2).

Additionally, twelve varieties (Bezostaja-1, Kate A-1, Ceyhan-99, Sönmez 2001, Sagittario, Vittorio, Aglika, Adelaide, Avorio, Bora, Boldane, and Hüseyinbey) were moderately resistant to, *PSTr-28*, *PSTr-29*, and *PSTr-30*, while only one variety, Rumeli, showed moderately resistant reaction to the races *PSTr-28*, *PSTr-30*, and *PSTr-31* (Table 2). On the other hand, 31 varieties (44% of the tested varieties) alarmingly showed susceptible reactions to all *Pst* races tested in this study.

Additionally, the correlation analysis of the reactions of the varieties to the *Pst* races showed that the strongest correlations were between *PSTr-28* and *PSTr-30* (*r* = 0.770, *p* = 0.000), *PSTr-28* and *PSTr-29* (*r* = 0.727, *p* = 0.000), and *PSTr-29* and *PSTr-30* (*r* = 0.721, *p* = 0.000), respectively (Table 4).

On the other hand, significant but weak correlations were found between *Yr5*-virulent race, *PSTr-27*, and other races, revealing that *PSTr-27* differs significantly from other races in terms of virulence on the varieties (Table 4).

### 3.2. Molecular Detection of Yr Genes

The presence and absence of nine resistant alleles of *Yr* genes, *Yr5*, *Yr10*, *Yr15*, *Yr17*, *Yr18*, *Yr26*, *Yr36*, *Yr44*, and *YrSP*, in the 70 bread wheat varieties were detected with different molecular markers (Appendix A). Sample agarose gel images for each molecular marker are given in Appendix A. The *Yr5*, *Yr15*, and *Yr26* genes were not detected in any of the varieties. The *Yr10* and *YrSP* genes were detected in only one variety each, namely Kıraç 66 and Adelaide, respectively. In addition, the *Yr17* gene was detected in three varieties, the *Yr18* gene in 26 varieties, the *Yr36* gene in 30 varieties, and the *Yr44* gene in 14 varieties (Appendix A).

No resistance genes were detected in 29 varieties, while 26 varieties were determined to contain two or more resistance genes. The varieties Ceyhan-99, Vittorio, Rumeli, Aglika, Gökkan, Avorio, Bora, and Genesi were found to carry three resistance genes (*Yr18*, *Yr36*, and *Yr44*) (Appendix A). The contribution of each *Yr* gene to resistance against *Pst* races was also estimated based on the ITs of the varieties. To this end, the ITs of varieties carrying the *Yr* gene(s) were compared with those lacking the *Yr* gene, using one-way ANOVA (Figure 2).

Remarkably, the lowest ITs were observed in varieties with the *Yr10 + 18* combination against all races. In addition, bread wheat varieties carrying the *Yr17* gene or the gene combinations *Yr18 + 36*, and *Yr18 + 36 + 44* had significantly lower ITs compared to those without the *Yr* gene (*Yr-*) against all *Pst* races (Figure 2).

## 4. Discussion

Plant diseases caused by many different plant pathogens pose an ongoing threat to global food security, as well as plant and ecosystem health. Plant pathogens are responsible for at least 20 to 40% of crop losses globally, and plant disease epidemics and outbreaks caused by these pathogens occur continuously every year, anywhere in the world. As an example of this, 617 new distribution records of 283 plant pathogens in 2021 and 29 pathogens with new distribution records in 2022 were reported [38,39]. For this reason, continued monitoring of these threats is essential to manage pathogen incursions and management of threats within a newly introduced area. In wheat, wheat rusts are the most devastating diseases historically worldwide, and many severe epidemics, causing serious economic damage, have occurred in many regions of the world, such as the stem rust epidemic, caused by the race *Ug99* [40], or the stripe rust epidemics, caused by *Yr9*-virulent races [25].

To date, more than 80 *Yr* genes (*Yr1*–*Yr83*) have been identified that confer resistance to this disease in wheat [16]. Among them, major resistance genes *Yr5* and *Yr15* have still been known to provide resistance to all *Pst* races worldwide. Notably, no virulent race has been reported globally against *Yr15*; however, several races have been reported as virulent to *Yr5,* especially during the last five years [6,17,20]. One of them, *PSTr-27*, was detected in Türkiye by our research group [6,17]. The other prevalent *Pst* races in Türkiye—*PSTr-28*, *PSTr-29*, *PSTr-30*, and *PSTr-31*—were also reported in these studies. The results of the present study showed that *Yr5*-virulent race, *PSTr-27*, was determined to be the most aggressive among the *Pst* races. Based on the aggressiveness of *Pst* races, *PSTr-27* was followed by *PSTr-31*, *PSTr-29*, *PSTr-28* and *PSTr-30*, respectively (Appendix A; Figure 1). Similarly, Zhang et al. [41] compared the relative parasitic fitness of the *Yr5*-virulent races (TSA-6 and TSA-9) identified in China [20] with four prevalent Chinese races (CYR31, CYR32, CYR33, and CYR34), and determined that *Yr5*-virulent races had significantly (*p* < 0.05) higher parasitic fitness than other *Pst* races, following the CYR32.

In the present study, seven varieties, Kıraç 66, İkizce 96, Ziyabey 98, Bayraktar 2000, Altındane, Dinç, and Shiro, showed a moderately resistant reaction to the most aggressive race, PSTr-27. Two of them, Kıraç 66 and Dinç, were also the most prominent varieties to all *Pst* races studied. Zhang et al. (2022) [26] evaluated the virulence and potential risk of two *Yr5*-virulent races (TSA-6 and TSA-9) identified in China in 2017, with other prevalent *Pst* races, on 165 Chinese wheat varieties. They determined that the avirulence/virulence patterns of these races were highly similar to the CYR32 race commonly found in China, but significantly different from CYR34. Of the 165 varieties, 21 showed all-stage resistance to the TSA-6 race, 34 to TSA-9, and 20 to both races. In both studies, about 10% of the varieties had a resistant reaction to *Yr5*-virulent races.

In addition to reaction tests, molecular screening of nine resistance genes (*Yr5*, *Yr10*, *Yr15*, *Yr17*, *Yr18*, *Yr26*, *Yr36*, *Yr44*, and *YrSP*) in bread wheat varieties was also investigated in this study. In addition to *Yr5* and *Yr15*, other resistance genes utilized in this study, including *Yr10*, *Yr17*, *Yr18*, *Yr26*, *Yr36*, *Yr44*, and *YrSP*, have also been used in breeding programs for yellow rust resistance in many regions of the world [42,43,44,45,46]. There are no varieties possessing the resistance genes *Yr5*, *Yr15*, and *Yr26* in these germplasms. Baloch et al. [47] screened a germplasm including Kazakh, Russian, and Turkish wheat varieties as well as Turkish wild emmer genotypes, and found no wheat varieties with *Yr15* gene except for Turkish wild emmer genotypes. Abbas et al. [48] also screened 349 Pakistan and Southwest China wheat genotypes for 13 major *Yr* genes, and determined that *Yr26*, *Yr15*, and *Yr65* were present with a higher percentage than other genes studied in both Pakistani and Chinese genotypes. In another study, Ul Islam et al. [49] evaluated 192 wheat genotypes for resistance to stripe rust, and reported that 9, 12, 14, and 32 of them had the resistance genes *Yr5*, *Yr10*, *Yr15*, and *Yr17*, respectively. Recently, Sucur et al. [50] evaluated European bread wheat varieties for resistance to stripe rust and powdery mildew and reported that *Yr5* and *Yr15* were remarkably high in these varieties. These results show that there is a wide variation in wheat germplasm for *Yr* genes.

In the present study, the relative effect of resistance genes on disease severity was also evaluated. According to the findings, as a single gene effect, only *Yr17* provided significant resistance to PSTr-28, PSTr-29 and PSTr-30 (Figure 2). However, some gene combinations were generally effective for resistance to the *Pst* races. Specifically, the combination *Yr10* + *Yr18* was prominent compared to the effects of other combinations, since it provided resistance to all *Pst* races studied. In addition, *Yr36* + *YrSP* and *Yr16* + *Yr36* + *Yr44* were other effective combinations, especially for resistance to PSTr-28, PSTr-29 and PSTr-30. In a similar way, Abbas et al. [48] reported that some gene combinations, such as *Yr26* + *Yr48*, *Yr29* + *Yr5*, *Yr26* + *Yr30*, and *Yr30* + *Yr17*, enhanced resistance to the currently prevalent *Pst* races in Pakistan. Leharwan et al. [51] claimed that the resistance genes, *Yr10*, *Yr15*, *Yr18*, *Yr24*, *Yr29*, *Yr36*, *Yr44*, *Yr53*, and *Yr65*, provided significant resistance to prevalent *Pst* races in India. Kokhmetova et al. [52] also reported that wheat genotypes carrying *Yr10* alone or in combination with other *Yr* genes enhanced resistance significantly to Kazakhstan *Pst* population. These results suggest that the effective *Yr* gene or gene combinations may vary depending on the *Pst* races commonly found in related region(s).

## 5. Conclusions

In the present study, according to reaction test results, the most aggressive races were determined to be as follows: *PSTr-27* > *PSTr-31* > *PSTr-28* > *PSTr-29* ≥ *PSTr-30*. Seven varieties (Kıraç 66, İkizce 96, Dinç, Altındane, Ziyabey 98, Bayraktar 2000, and Shiro) gave a moderately resistant reaction to *Yr5*-virulent, *PSTr-27*. However, the remaining 63 varieties, which are widely cultivated in wheat-growing areas of Türkiye, were susceptible to this *Pst* race. The most interesting point here is that most of the resistant varieties to *PSTr-27* do not contain any *Yr* gene(s) studied, according to molecular screening. The same pattern was observed in the reactions of the varieties to other races; overall, 19 varieties without resistance genes studied to all *Pst* races were susceptible, whereas 10 varieties had resistance to one or more races, suggesting the presence of unknown or novel resistance sources. The results show that multiple genes, like the *Yr10* + *Yr18* combination, can enhance resistance to *Pst* races, and this suggests that more genes should be used in wheat resistance breeding programs to achieve durable resistance to *Pst*. In addition, the *Yr5*-virulent race *PSTr-27*, which had previously been detected by our research group from only a single isolate and was not widespread across Türkiye, was found in this study to exhibit high virulence on widely grown wheat varieties; therefore, continual monitoring of *PSTr-27* and other *Pst* races is essential.

## Figures and Tables

**Figure 1 jof-11-00635-f001:**
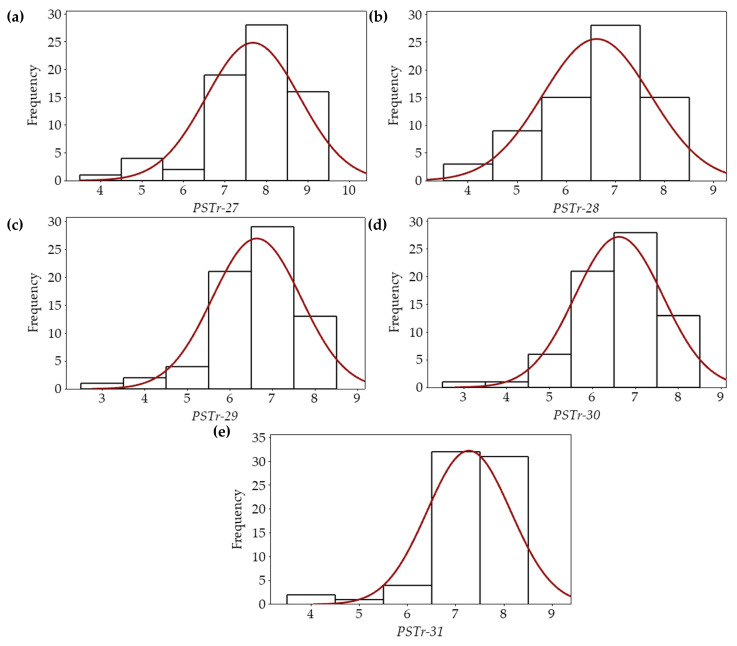
Histograms showing the reactions of bread wheat varieties to the *PSTr-27* (**a**), *PSTr-28* (**b**), *PSTr-29* (**c**), *PSTr-30* (**d**), and *PSTr-31* (**e**).

**Figure 2 jof-11-00635-f002:**
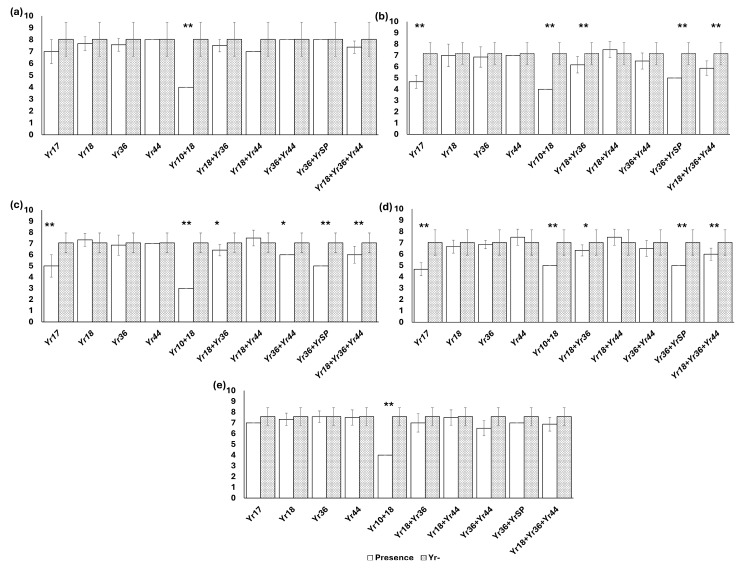
Contribution of each Yr gene or gene combination for resistance to *PSTr-27* (**a**), *PSTr-28* (**b**), *PSTr-29* (**c**), *PSTr-30* (**d**), and *PSTr-31* (**e**). Each bar represents the average ITs of bread wheat varieties with presence or absence (*Yr-*) of *Yr* gene(s). * *p* < 0.05, ** *p* < 0.01.

**Table 1 jof-11-00635-t001:** The molecular markers used in this study for detection of each *Yr* gene.

Gene	Marker Type	Marker	Annealing Temp (°C)	Sequence (5′→3′)	Expected Fragment (bp)	Reference
*Yr5*	STS/CAPS	*STS7*	45	GTACAATTCACCTAGAGT	289 (+)	[29]
		*STS8*		GCAAGTTTTCTCCCTAT	182 (−)	
*Yr10*	Gene-specific	*Yr 10 F*	64	TCAAAGACATCAAGAGCCGC	543 (+)	[30]
		*Yr 10 R*		TGGCCTACATGAACTCTGGAT		
*Yr15*	SSR	*Xbarc8 F*	50	GCGGGGGCGAAACATACACATAAAAACA	200 (+)	[31]
		*Xbarc8 R*		GCGGGAATCATGCATAGGAAAACAGAA	280 (−)	
*Yr17*	SCAR	*VENTRIUP*	65	AGGGGCTACTGACCAAGGCT	262 (+)	[32]
		*LN2*		TGCAGCTACAGCAGTATGTACACAAAA		
*Yr18*	Gene-specific	*L34DINT9F*	51	TTGATGAAACCAGTTTTTTTTCTA	517 (+)	[33]
		*L34PLUSR*		GCCATTTAACATAATCATGATGGA		
*Yr26*	STS	*we173*	57	GGGACAAGGGGAGTTGAAGC	451/500 (+)	[34]
				GAGAGTTCCAAGCAGAACAC	730 (−)	
*Yr36*	Gene-specific	*Yr36START*	52	GGCCACACTGCAATACTATACC	871 (+)	[35]
				CACAAATCCTGGCTGTGGAC		
*Yr44*	STS	*pWB5*	49	GGTGCAATTTGAGTTTGGAGT	380 (+)	[36]
		*pWN1R1*		GGTGTTGACTGGAGAATCCG		
*YrSP*	STS	*dp269*	55	CTGCTGTCACCGCTCTCC	190 (+)	[37]
				AGTCACACGCCCTACTCTCC	201 (−)	

**Table 2 jof-11-00635-t002:** Reactions of bread wheat varieties to *Pst* races.

Variety	*Pst* Races				
	*PSTr-27*	*PSTr-28*	*PSTr-29*	*PSTr-30*	*PSTr-31*
Bezostaja-1	S	MR	MR	MR	S
Kıraç 66	MR	MR	R	MR	MR
Cumhuriyet 75	S	S	S	S	S
Gerek 79	S	S	S	S	S
Atay-85	S	S	S	S	S
Kate A-1	S	MR	MR	MR	S
Dağdaş 94	S	MR	MR	MR	MR
Sultan 95	S	S	S	MR	S
Kaşifbey 95	S	MR	S	MR	S
İkizce 96	MR	S	S	S	S
Pamukova 97	S	S	S	S	S
Pehlivan	S	S	MR	S	S
Karacadağ 98	S	S	S	S	S
Gönen 98	S	MR	MR	MR	MR
Ziyabey 98	MR	MR	MR	MR	S
Karahan-99	S	MR	S	MR	S
Ceyhan-99	S	MR	MR	MR	S
Flamura 85	S	MR	S	S	S
Bayraktar 2000	MR	MR	MR	MR	S
Demir 2000	S	S	S	S	S
Sönmez 2001	S	MR	MR	MR	S
Alparslan	S	MR	MR	S	S
Pandas	S	S	S	S	S
Sagittario	S	MR	MR	MR	S
Canik 2003	S	S	S	S	S
Tosunbey	S	S	MR	MR	S
Ahmetağa	S	S	S	S	S
Krasunia odes’ka	S	S	S	S	S
Kenanbey	S	S	MR	S	S
Aldane	S	S	S	S	S
Selimiye	S	MR	S	MR	S
ES 26	S	S	MR	S	S
Esperia	S	S	MR	S	MR
Cömert	S	S	S	S	S
Altındane	MR	S	S	S	S
Vittorio	S	MR	MR	MR	S
Quality	S	S	S	S	S
Rumeli	S	MR	S	MR	MR
Aglika	S	MR	MR	MR	S
Dinç	MR	MR	MR	R	MR
Gökkan	S	S	S	S	S
Segor	S	S	S	S	S
Adelaide	S	MR	MR	MR	S
Avorio	S	MR	MR	MR	S
Tekin	S	S	S	S	S
Nevzatbey	S	S	MR	S	S
Yakamoz	S	S	S	S	S
Bora	S	MR	MR	MR	S
Genesi	S	MR	MR	MR	MR
Glosa	S	S	S	S	S
Masaccio	S	S	S	S	S
Efe	S	S	MR	MR	S
Kale	S	S	S	MR	S
Yüksel	S	S	S	S	S
Leuta	S	S	S	S	S
Duru	S	S	S	S	S
Hüseyinbey	S	MR	MR	MR	S
Albachiara	S	S	S	S	S
Damla	S	S	S	S	S
Koç	S	MR	S	S	S
İzvor	S	S	S	S	S
Anafarta	S	S	S	S	S
Abide	S	S	S	S	S
Eylül	S	S	S	S	S
Albaşak	S	S	S	S	S
Boldane	S	MR	MR	MR	S
Beyaz-I	S	S	S	S	S
Shiro	MR	MR	MR	MR	S
Destra	S	S	S	S	S
Alba	S	S	S	MR	S

**Table 3 jof-11-00635-t003:** Number of varieties showing resistant or moderately resistant reactions to one or more *Pst* races.

No of Variety	*Pst* Races				
	*PSTr-27*	*PSTr-28*	*PSTr-29*	*PSTr-30*	*PSTr-31*
2	+	−	−	−	−
2	−	+	−	−	−
4	−	−	+	−	−
3	−	−	−	+	−
1	−	+	+	−	−
3	−	+	−	+	−
2	−	−	+	+	−
1	−	−	+	−	+
12	−	+	+	+	−
1	−	+	−	+	+
3	+	+	+	+	−
3	−	+	+	+	+
2	+	+	+	+	+
31	−	−	−	−	−
Total	7	27	28	29	7

**Table 4 jof-11-00635-t004:** Correlation coefficients between five *Pst* races based on the ITs of the varieties.

Race	*PSTr-27*	*PSTr-28*	*PSTr-29*	*PSTr-30*	*PSTr-31*
*PSTr-27*	1				
*PSTr-28*	0.519 **	1			
*PSTr-29*	0.527 **	0.727 **	1		
*PSTr-30*	0.516 **	0.770 **	0.721 **	1	
*PSTr-31*	0.464 **	0.601 **	0.581 **	0.543 **	1

** *p* < 0.01.

## Data Availability

The data presented in this study is available on request from the corresponding author.

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
