# Peer review of "Evaluation of the Potential Risk Posed by Emerging Yr5-Virulent and Predominant Races of Puccinia striiformis f. sp. tritici on Bread Wheat (Triticum aestivum L.) Varieties Grown in Türkiye"

_jof, 2025, doi:10.3390/jof11090635_

Round 1

Reviewer 1 Report

The manuscript written by Akan et al. reported nine important resistance genes in wheat (Triticum L.) against the Bread Wheat caused by the Puccinia striiformis f. sp. tritici using sampling from Türkiye. It is a subject of interest for global agriculture and I am in favor of publishing it, since the article is well written and provides a variety of information about these genes, which may be useful in the literature for comparison with samples from other countries.

Suggestions: 

- Increase the font size of the graph axes and of legends in Figures 1 and 2;

- Transfer the Figure S1 of the Material Supplementar to the manuscript as Figure 1.

Author Response

Dear reviewer,

First, we would like to thank you for your valuable contributions. We re-checked the manuscript in detail and revised the figures as suggested.

Comments 1: Increase the font size of the graph axes and of legends in Figures 1 and 2

Response 1: We agree with this comment, and so we revised the axes and legends of the figures.

Comments 2: Transfer the Figure S1 of the Material Supplementar to the manuscript as Figure 1.

Response 2: We partially agree with this suggestion, but the relevant figure has been included in the supplementary for evidence purposes only. Therefore, we believe it would be more appropriate to include it in the supplement.

Reviewer 2 Report

The study possesses both scientific and practical significance: for the first time in Türkiye, the reactions of 70 bread-wheat varieties have been systematically evaluated against five dominant races of Puccinia striiformis f. sp. tritici (Pst), including the highly aggressive PSTr-27 race that is virulent on the Yr5 gene. For global relevance, it would be valuable to extend this assessment to wheat varieties cultivated on a continental scale. The findings provide a solid foundation for breeding new varieties that combine known genes (Yr10, Yr18) with potentially novel Yr genes, thereby reducing the likelihood of future epiphytotics.

Recommendations

Is it possible to conduct resistance assessments in field conditions with natural infection over a period of 3–5 years?

2.2. Multiplication of urediniospores of Pst and reaction tests of the varieties 

- The source of the spores used for the primary inoculation is not specified, nor is the origin of the strains. The concentration of spores should be precisely calculated in spores/ml rather than mg/ml.

- Growing Conditions: The temperature range is too broad; an optimal range of 12–14°C is necessary for stable rust development. Additionally, humidity should be reduced to 60–70% after inoculation to avoid fungal contamination.

- Recommendations for Further Research: The assessment of resistance was conducted using the McNeal scale (0–9), which is a visual scale for evaluating the degree of leaf infection in wheat by yellow rust (Puccinia striiformis). However, the 0–4 scale developed by Roelfs et al. is a more modern and accurate system for assessing the level of leaf rust infection. Unlike the 0–9 McNeal scale, it simplifies classification and reduces subjectivity in visual assessments.

- In my opinion, conducting the assessment at 19 days post-inoculation is too early; the visualization of disease symptoms typically occurs between days 21 and 24.

2.3. Extraction of genomic DNA

Please specify the method used for leaf homogenization (e.g., freezing in liquid nitrogen, ball mill, etc.).

2.4. Molecular detection of Yr genes

Provide validation using an alternative method.

3.2. Molecular detection of Yr genes

Clarify the details of the screening process by including the markers used for each gene along with their key characteristics (functionality, marker type—SNP, STS, etc.). This information is essential for assessing the conclusions and comparing them with other studies.

5. Conclusions

Section 5: Clarify why the majority of resistant wheat cultivars do not contain the studied Yr genes.

Author Response

Dear reviewer,

First, we would like to thank you for your valuable contributions. We re-checked the manuscript in detail and revised as suggested.

Comments 1: The source of the spores used for the primary inoculation is not specified, nor is the origin of the strains. The concentration of spores should be precisely calculated in spores/ml rather than mg/ml.

Response 1: We agree with you; however, the source of the spores and the origin of the races have already been cited in a previously published article (Cat et al., 2023) by our research group. In addition, we partially agree with your comment regarding concentration, since in all studies conducted in this field the concentration is reported in mg/ml. Examples of such studies are provided below.

Cat, A.; Tekin, M.; Akan, K.; Akar, T.; Catal, M. Virulence characterization of the wheat stripe rust pathogen, Puccinia striiformis f. sp. tritici, in Turkey from 2018 to 2020. Can J Plant Pathol 2023, 45, 158-167.

Sorensen, C.K.; Thach, T.; Hovmoller, M.S. Evaluation of spray and point inoculation methods for the phenotyping of Puccinia striiformis on wheat. Plant Dis 2016, 100, 1064-1070.

Zhang, G.S.; Liu, W.; Wang, L.; Cheng, X.R.; Tian, X.X.; Du, Z.M.; Kang, Z.S.; Zhao, J. Evaluation of the potential risk of the emerging virulent races of Puccinia striiformis f. sp. to 165 Chinese wheat cultivars. Plant Dis 2022, 106, 1867-1874.

Comments 2: The temperature range is too broad; an optimal range of 12–14°C is necessary for stable rust development. Additionally, humidity should be reduced to 60–70% after inoculation to avoid fungal contamination.

Response 2: We do not agree with you, especially for temperature. We consider that 10-16°C is appropriate for Pst development. Similar temperature ranges were also reported in the articles given in the previous response. However, we agree with you for the comment about humidity. We already used the 60-70% humidity during incubation, and so revised the sentence.

Comments 3: The assessment of resistance was conducted using the McNeal scale (0–9), which is a visual scale for evaluating the degree of leaf infection in wheat by yellow rust (Puccinia striiformis). However, the 0–4 scale developed by Roelfs et al. is a more modern and accurate system for assessing the level of leaf rust infection. Unlike the 0–9 McNeal scale, it simplifies classification and reduces subjectivity in visual assessments.

Response 3: Thank you for your suggestion; we will take it into consideration for future studies.

Comments 4: In my opinion, conducting the assessment at 19 days post-inoculation is too early; the visualization of disease symptoms typically occurs between days 21 and 24.

Response 4: This is an observation-related issue, and in rust studies observations are generally made 19 days after inoculation. Therefore, we used the term ‘approximately’ in the manuscript.

Comments 5: Please specify the method used for leaf homogenization (e.g., freezing in liquid nitrogen, ball mill, etc.)

Response 5: Specified and revised the sentence in detail.

Comments 6: Provide validation using an alternative method. Clarify the details of the screening process by including the markers used for each gene along with their key characteristics (functionality, marker type—SNP, STS, etc.). This information is essential for assessing the conclusions and comparing them with other studies.

Response 6: The relevant molecular markers have also been used in previous similar studies, and the publication details are provided below. In addition, the information regarding the markers used has already been presented in Table 1.

Comments 7: Clarify why the majority of resistant wheat cultivars do not contain the studied Yr genes.

Response 7: The conclusion section was revised as suggested.

Reviewer 3 Report

The manuscript evaluates potential risks posed by emerging and predominant races of Puccinia striiformis f. sp. tritici (Pst) on bread wheat varieties in Türkiye.  The authors suggested that the combination of Yr10 and Yr18 provided resistance to all of the studied races, and Yr5-virulent PSTr-27b was the most aggressive race. They also found that most of the resistant varieties didn’t have the nine resistant genes that were analyzed. They used nine molecular markers to detect the presence of Yr-resistant alleles against Pst. The manuscript would benefit from a thorough revision to improve its clarity and quality.  There are several areas where the English could be improved for clarity and flow. A language edit is recommended.  Although the study examined nine Yr genes, the lack of resistance in many wheat varieties indicates that other Yr genes or quantitative trait loci (QTLs) might also play a role. Broadening the genetic analysis could offer a more comprehensive understanding. The research focuses solely on wheat varieties cultivated in Türkiye, which aligns with the study’s goals, but the results may not be directly transferable to other regions without additional validation.

My comments are as follows:

Abstract

Please rewrite the abstract

Line 23: Define ‘reaction tests’

Line 27: AT the molecular level instead of ‘molecularly’

Lines 30-32: Nine Yr genes instead of ‘the studied Yr genes’

Please mention how many resistant varieties didn’t carry nine resistant genes.

Line 40: Was instead of ‘is’

Line 47: Wheat-growing area is not susceptible

Line 55: Epidemics lead to

Line 58: ineffective against

Line 59: Differential lines

Lines 62-69: Please revise

Line 71: Confer instead of ‘are known as’

Line 71-73: Please revise

Line 75: remove ‘strikingly’

Line 75: China, Turkey (Turkiye), and Syria are 3 countries you can’t use ‘both’

Line 77-82: Please revise. For example, therefore, instead of ‘For this reason’ and host and/or intermediate host

Line 90: Remove ‘China’

Line 93-95: Please revise

Materials and methods

Line 101: The wheat variety Morocco

Line 105-112: Present the content in a table and explain this clearly

Line 114: Pst urediniospores instead of ‘uredinispores of Pst

Line 116: Check variety Morocco

Line 118: Remove ‘of the variety’

Line 126: remove ‘daily’

Line 126: Start the sentence with, Each pot was enclosed

Line 127: Remove ‘among each other’

Line 131: Remove ‘for each Pst race’

Line 132: Conducted on

Line 133-134: Revise the sentence and add a reference

Line 144-146: Please revise

Line 144: How old were the plants?

Line 145: Yr gene line?

Line 148: Remove ‘Each’

Line 150: Remove ‘Later’, DNA was instead of ‘DNAs were’, and what was the pH of the buffer?

Line 154-156: Please revise

Line 157: Yr gene lines?

Line 166: What was the pH of the buffer

Line 168: Please specify what DNA standard was used

Line 172: Remove ‘The information about’

Line 175-177: Please revise

Line 180: p-value is italicized and lower case

Results

Lines 188-192: Is redundant

Line 192-194: Please revise

Line 198: ‘many varieties’, please specify, and the sentence is not completed

Lines 203-205: Redundant

Lines 203-216: Please revise

Line 221: Remove ‘all’

Line 224: Remove ‘it was determined that’

Line 225: displayed instead of ‘have’

Line 228: Remove ‘determined’

Line 234: Remove ‘found’

Line 236: Remove ‘the races’

Line 240: remove ‘were determined’

Line 251: Resistant alleles of Yr genes

Line 254: Remove ‘According to the molecular findings’

Line 255-256: Please revise

Line 262-264: It is not clear

Discussion

Line 275-282: This paragraph is general information, not relevant, and should be removed

Line 286-288: It is repetitive, please remove

Line 289: That confer instead of ‘as conferring’

Line 291: provide resistance to all

Line 293: Please italicize PSTr-27

Lines 293-295: Please revise

Lines 296-300: The content is repetitive

Line 321: Remove ‘molecularly’ and ‘set’

Line 322: Remove ‘they’

Line 325: Remove ‘remarkably’

Line 343: significant instead of ‘significantly’

References

The references don’t follow a consistent format. Please ensure that the format meets the guidelines set by the journal, e.g., italicized versus non-italicized words, full names of journals versus abbreviated names, boldface versus non-bold numbers, etc.

Author Response

Dear reviewer,

First, we would like to thank you for your valuable contributions. We re-checked the manuscript in detail and revised as suggested.

Comments 1: Line 23: Define ‘reaction tests’

Response 1: We revised the start of the sentence as “Reaction tests of wheat varieties to all races”.

Comments 2: Line 27: AT the molecular level instead of ‘molecularly’

Response 2: We revised it as at the molecular level.

Comments 3: Lines 30-32: Nine Yr genes instead of ‘the studied Yr genes’. Please mention how many resistant varieties didn’t carry nine resistant genes.

Response 3: We revised the sentence as suggested.

Comments 4: Line 40: Was instead of ‘is’

Response 4: revised

Comments 5: Line 55: Epidemics lead to

Response 5: revised

Comments 5: Line 58: ineffective against

Response 5: revised

Comments 6: Line 59: Differential lines

Response 6: revised

Comments 7: Lines 62-69: Please revise

Response 7: revised

Comments 8: Line 71: Confer instead of ‘are known as’

Response 8: revised

Comments 9: Line 71-73: Please revise

Response 9: revised the sentences in detail.

Comments 10: Line 75: remove ‘strikingly’

Response 10: removed

Comments 11: Line 75: China, Turkey (Turkiye), and Syria are 3 countries you can’t use ‘both’

Response 11: removed

Comments 12: Line 77-82: Please revise. For example, therefore, instead of ‘For this reason’ and host and/or intermediate host

Response 12: revised the sentences in detail.

Comments 13: Line 90: Remove ‘China’

Response 13: removed

Comments 14: Line 93-95: Please revise

Response 14: revised the sentences in detail.

Comments 15: Line 101: The wheat variety Morocco

Response 15: revised

Comments 16: Line 105-112: Present the content in a table and explain this clearly

Response 16: Given the large number of tables in the manuscript, we believe that presenting it in this way is more appropriate.

Comments 17: Line 114: Pst urediniospores instead of ‘uredinispores of Pst

Response 17: revised

Comments 18: Line 116: Check variety Morocco

Response 18: revised

Comments 19: Line 118: Remove ‘of the variety’

Response 19: removed

Comments 20: Line 126: remove ‘daily’

Response 20: removed

Comments 21: Line 126: Start the sentence with, Each pot was enclosed

Response 21: revised

Comments 22: Line 127: Remove ‘among each other’

Response 22: removed

Comments 23: Line 131: Remove ‘for each Pst race’

Response 23: removed

Comments 24: Line 132: Conducted on

Response 24: revised

Comments 25: Line 133-134: Revise the sentence and add a reference

Response 25: revised in detail

Comments 26: Line 144-146: Please revise

Response 26: revised in detail

Comments 27: Line 144: How old were the plants?

Response 27: added

Comments 28: Line 145: Yr gene line?

Response 28: revised

Comments 29: Line 148: Remove ‘Each’

Response 29: removed

Comments 30: Line 150: Remove ‘Later’, DNA was instead of ‘DNAs were’, and what was the pH of the buffer?

Response 30: revised and added information about pH.

Comments 31: Line 154-156: Please revise

Response 31: revised

Comments 32: Line 157: Yr gene lines?

Response 32: revised

Comments 33: Line 166: What was the pH of the buffer

Response 33: added

Comments 34: Line 168: Please specify what DNA standard was used

Response 34: specified

Comments 35: Line 172: Remove ‘The information about’

Response 35: removed

Comments 36: Line 175-177: Please revise

Response 36: revised

Comments 37: Line 180: p-value is italicized and lower case

Response 37: revised

Comments 38: Lines 188-192: Is redundant

Response 38: We believe that an explanatory introduction is necessary.

Comments 39: Line 192-194: Please revise

Response 39: revised

Comments 40: Line 198: ‘many varieties’, please specify, and the sentence is not completed

Response 40: revised the sentence

Comments 41: Lines 203-205: Redundant; Comments 42: Lines 203-216: Please revise

Response 41 and 42: revised in detail

Comments 43: Line 221: Remove ‘all’

Response 43: removed

Comments 44: Line 224: Remove ‘it was determined that’

Response 44: removed

Comments 45: Line 225: displayed instead of ‘have’

Response 45: revised

Comments 46: Line 228: Remove ‘determined’

Response 46: removed

Comments 47: Line 234: Remove ‘found’

Response 47: removed

Comments 48: Line 236: Remove ‘the races’

Response 48: removed

Comments 49: Line 240: remove ‘were determined’

Response 49: removed

Comments 50: Line 251: Resistant alleles of Yr genes

Response 50: revised

Comments 51: Line 254: Remove ‘According to the molecular findings’

Response 51: removed

Comments 52: Line 255-256: Please revise

Response 52: revised

Comments 53: Line 262-264: It is not clear

Response 53: revised

Comments 54: Line 275-282: This paragraph is general information, not relevant, and should be removed

Response 54: Since this part serves as a linking introduction, we think it would be useful to keep the introduction section for clarity and context.

Comments 55: Line 286-288: It is repetitive, please remove

Response 55: removed

Comments 56: Line 289: That confer instead of ‘as conferring’

Response 56: revised

Comments 57: Line 291: provide resistance to all

Response 57: revised

Comments 58: Line 293: Please italicize PSTr-27

Response 58: corrected

Comments 59: Lines 293-295: Please revise

Response 59: revised

Comments 60: Lines 296-300: The content is repetitive

Response 60: removed

Comments 61: Line 321: Remove ‘molecularly’ and ‘set’

Response 61: removed

Comments 62: Line 322: Remove ‘they’

Response 62: removed

Comments 63: Line 325: Remove ‘remarkably’

Response 63: removed

Comments 64: Line 343: significant instead of ‘significantly’

Response 64: corrected

Comments 65: The references don’t follow a consistent format. Please ensure that the format meets the guidelines set by the journal, e.g., italicized versus non-italicized words, full names of journals versus abbreviated names, boldface versus non-bold numbers, etc.

Response 65: We used the software EndNote to generate the reference list; however, it seems that some errors occurred in the publication details. All of them have now been corrected.

Round 2

Reviewer 3 Report

The authors have carefully addressed the comments and revised the manuscript accordingly.

The authors have carefully addressed the comments and revised the manuscript accordingly.